# Policy Transfer for Improved Sample Efficiency in Goal-Conditioned Reinforcement Learning

## Abstract

Goal-Conditioned Reinforcement Learning (GCRL) tackles the challenging problem of long-horizon, sparse-reward goal-reaching tasks with continuous actions. Recent methods, relying on a two-level hierarchical policy along with a graph of sub-goal landmarks, have demonstrated reasonable asymptotic performance. However, existing algorithms suffer from poor sample efficiency due to excessively many uninformative landmarks and the inability to transfer low-level behaviour between related tasks. We instead claim that transferring a pre-trained low-level policy between environments can improve landmark generation and dramatically improve sample efficiency and even success rates. We introduce an algorithm PROMO, which explicitly models reachability using a pre-trained low-level policy and uses it to improve landmark generation and transfer low-level behaviour. We demonstrate 3-4x improvements in sample efficiency over existing state-of-the-art methods on the challenging robotics tasks of AntMaze and Reacher3D, with the mild overhead of one-time policy pre-training. In addition, our method achieves superior success rates across all environments, as well as better training stability and much fewer, more informative landmarks.

## 1 Introduction

GCRL is a promising paradigm to solve Reinforcement Learning (RL) tasks parameterised by a desired goal. These tasks can encode far more challenging information than their standard RL counterparts, owing to the parameter-dependent, sparse reward structure. Many important spatial applications such as robotic navigation and manipulation can be expressed as goal-reaching tasks in low-dimensional goal spaces. Recently, several methods have tackled this problem by building a graph of *landmarks*, i.e. important sub-goals which facilitate goal space coverage. These methods employ a hierarchical structure, with a low-level policy implementing sub-goals and a high-level policy maintaining the graph and choosing sub-goals for the low-level to implement. Shortest-path algorithms are used to navigate through the graph. While these techniques have proven worthwhile, they are still inefficient in terms of environment steps needed for training.

GCRL trains the separate skills of *moving* and *planning* together, potentially allowing for a more expressive overall policy, since the two levels communicate and develop together. However, this means that the skill of moving through the goal space must be re-learned for every new environment, leading to very large sample complexity. Conversely, skills-based Hierarchical RL methods (Sohn et al., 2018; Li et al., 2019; Tessler et al., 2017; Frans et al., 2018) seek to pre-train multiple "primitive" skills (policies) and then transfer them to more complex environments, but these methods do not make use of the strong prior of goal-reaching rewards.

In this work, we seek to bridge this gap by studying whether transferable skills can improve sample efficiency in goal-conditioned problems by facilitating better balance between exploration (novelty) and exploitation (reachability). This may open new avenues for the use of transfer in RL. Our main contributions are:

- To our knowledge, the first application of pre-trained transferable skills in a GCRL context

- A new reachability modelling method based on the transferred policy, along with a new landmark generation procedure which utilises the reachability model to produce fewer, more spaced out landmarks than existing methods.

- An extensive set of comparative, ablative and investigative experiments demonstrating the ability of transfer-based reachability modelling to greatly improve both sample efficiency and goal-reaching success over state-of-the-art methods.

## 2 RELATED WORK

Hierarchical GCRL methods follow the feudal structure of a high-level policy providing sub-goals to a low-level policy, with both policies being trained together (Vezhnevets et al., 2017). Some early examples include HAC (Levy et al., 2017), HIRO (Nachum et al., 2018) and RIS (Chane-Sane et al., 2021), where the high-level policy directly outputs a subgoal. Other methods build a graph of static landmarks (important sub-goals) and use it to plan over long distances (Zhang et al., 2021; Kim et al., 2021; Huang et al., 2019) using a shortest path algorithm, thereby improving efficiency. The graph nodes are landmarks, and the edges between them depend on some measure of similarity or reachability. Search on the Replay Buffer (SoRB) (Eysenbach et al., 2019) takes all states from the low-level replay buffer as landmarks and constructs a graph based on them. While these methods achieved some early success, they still suffer from poor sample efficiency and many uninformative landmarks. Moreover, rather than a fixed or constrained initial state distribution (the most challenging case), they rely on the distribution uniformly covering the full space while training, something that is impractical in many applications.

The current state-of-the-art methods, DHRL (Lee et al., 2022), BEAG (Yoon et al., 2024) and NGTE (Park et al., 2024) are also landmark/graph-based and improved performance on the more challenging fixed distribution case, but still suffer from poor sample efficiency and excessive landmarks. DHRL samples landmarks as achieved goals from the low-level replay buffer and focuses on decoupling the low-level and high-level time horizons, while BEAG assigns landmarks as the vertices of an adaptively refined grid. Both these methods produce many uninformative landmarks. NGTE navigates to a frontier landmark and uses it as an exploration outpost, similar to ours, but then generates landmarks randomly.

All the above methods implicitly or explicitly balance reachability with novelty when generating landmarks/subgoals. However, these quantities are represented by overly simplistic heuristics. For example, many works use the low-level value function to assess reachability (e.g. DHRL, SoRB, RIS), which is only accurate for close-by states (it is trained with one-step temporal difference errors between close-by states). Other methods use the empirical test of whether a given landmark *actually was* reached during exploration, without any modelling. NGTE does this on randomly sampled landmarks and BEAG makes the grid more coarse (novelty-seeking) or fine (reachability-seeking) based on whether landmarks were reached.

By contrast, we explicitly model reachability between all (including far-apart) possible sub-goals, using a pre-trained and transferred low-level policy and a novel trajectory-accessibility architecture. Novelty is measured as non-reachability from the existing landmarks. Our landmark generation procedure can therefore balance reachability and novelty of very distant goals to generate much fewer, more spaced out and therefore more informative landmarks. In addition, our method is the first to pre-train and transfer the low-level policy to help in landmark generation and reuse the *moving* behaviour across related tasks.

## 3 PRELIMINARIES

**Goal-augmented MDP.** We formulate the problem as a finite-horizon Goal-augmented Markov Decision Process (GA-MDP) (Sutton & Barto, 2018; Nasiriany et al., 2019), defined by a 10-tuple $M = (S, A, G, T, R, d, \phi, \gamma, \rho_S, \rho_G)$. The sets $S$, $A$ and $G$ are the state, action and goal spaces respectively. We assume that G is bounded, a common assumption made by e.g. BEAG (Yoon et al., 2024). State, action and goal spaces are usually continuous and Euclidean in GCRL, though this is not a strict requirement. We adopt a convention of denoting *achieved* goals, i.e., the agent's current position in the goal space, by $g$ and *desired* goals, e.g. the task goal, by $h$. $T : S \times A \to S$ and

$R : S \times A \times G \to \mathbb{R}$ is a (possibly stochastic) dynamics transition function. Additionally, the known distance function $d(g_1, g_2) : G \times G \to \mathbb{R}^+$ (inducing the 2-norm if spaces are Euclidean) is used to determine reward. Specifically, the reward function $R : S \times A \times G \to \mathbb{R}$ is 0 if the achieved goal is within a known threshold $\delta_r \in \mathbb{R}^+$ of the desired goal, and $-1$ otherwise. This sparse reward case is the most challenging configuration, on which we focus in this work, but a dense reward can be defined as the negative distance to the goal. $\gamma \in [0, 1]$ is a discount factor and $\rho_S$ and $\rho_G$ are the initial state distribution and the task goal distribution. At the start of each episode, an initial state and a goal are sampled from these distributions. Finally, $\phi : S \to G$ is a known mapping from states to corresponding achieved goals, usually taken as the selection of the appropriate vector elements.

**Policy structure.** We endow the agent with a stochastic low-level policy $\pi_{\text{LL}}(a \,|\, s, h_{\text{LL}}) : A \times S \times G \to [0, 1]$ and a deterministic high-level planner $\pi_{\text{HL}}(h_{\text{LL}} \,|\, s, h) : G \times S \times G \to \{0, 1\}$, where $h_{\text{LL}}$ is a sub-goal passed periodically from $\pi_{\text{HL}}$ to $\pi_{\text{LL}}$, giving a primitive action $a$ for the environment.

Steps where new sub-goals are produced are called *control steps* and all steps in between them produce actions by passing the same sub-goal as at the most recent control step. The initial step of an episode is always a control step, but the rest are determined by the planner. The policies must be optimised to maximise the expected discounted *return* (discounted sum of rewards) over the task horizon.

**Accessibility and reachability.** Here we present two concepts which will play a central role throughout the paper.

**Definition** (Accessible goal space). We formally define the *accessible* goal space (or region) $G_{\text{acc}}$ of $M$ as the largest subset of $G$ such that for every $g \in G_{\text{acc}}$, there exists an initial state $s_1 \sim \rho_S$, a sequence $(s_2, \ldots, s_{N+1})$ of states generated by $T$, some action sequence $(a_1, \ldots, a_N)$ and some desired goal sequence $(h_1, \ldots, h_N)$, satisfying $\phi(s_{N+1}) = g$. This region is intuitively the set of points in goal space that can be reached at all (e.g. corridors as opposed to wall interior points in a maze). A point within this set is called *accessible*.

**Definition** (Reachable goal). For any two goals $g_1, g_2 \in G$, we say that $g_2$ is *reachable* from $g_1$ with respect to a policy $\pi(a \,|\, s, h)$ in $k$ steps, if the expected state sequence $(s_1, \ldots, s_{k+1})$ produced by $\pi$ under $T$, given any initial state $s_1$ for which $\phi(s_1) = g_1$, satisfies $d(\phi(s_{k+1}), g_2) < \delta_r$. This concept assesses the ability to move from one goal to another under the action of a given policy.

Finally, we define the functions $A(g) : G \to \{0, 1\}$ and $R_\pi(g_1, g_2) : G \times G \to \{0, 1\}$ to be indicator functions for accessibility and reachability, respectively. We will model these and use them for landmark generation.

## 4 METHOD

In this section, we present our algorithm **P**rogressive **R**eachability **O**ptimisation and **MO**delling (PROMO). PROMO consists of a low-level GCRL policy $\pi_{\text{LL}}$, pre-trained in a simpler environment, and a high-level graph-based planner $\pi_{\text{HL}}$, trained in the given complex environment using $\pi_{\text{LL}}$ as a fixed sub-policy. One episode in the low-level environment corresponds to one high-level control step in the main environment, though the planner may incite a new control step early if it chooses.

The training procedure has three steps: low-level policy training, trajectory model training and high-level planner training (including landmark generation). At inference time, the trajectory and accessibility models (together, the *reachability* model) are used to create a landmark graph at each control step, through which the planner finds the shortest path to the goal. Landmarks (or the end goal) are provided as subgoals to the low-level policy. The full inference flow is presented in Figure 2 while the low-level and high-level training scenarios are shown in Figure 1.

We distinguish between two formats for providing sub-goals to $\pi_{\text{LL}}$: *relative* and *absolute* goals. For translation-invariant tasks like maze navigation, sub-goals provided by $\pi_{\text{HL}}$ are relative to the achieved goal at the most recent control step, whereas they are the standard absolute goals for all other tasks. Navigation tasks are translation-invariant since the skill of moving a given displacement does not depend on the absolute starting position. The remainder of this section details the three-step training procedure and the architectural components. Note that Sections 4.5 (Reachability Model)

and 4.6 (Planner) are relevant for both inference time and exploration (navigating to frontier landmarks) during training time. Full pseudocode for the planner training (Algorithm 1), the exploration procedure (Algorithm 2) and the planner inference (Algorithm 3) can be found in the Appendix.

## 4.1 Step 1: Low-level Policy Training

The low-level policy, $\pi_{\mathrm{LL}}$, is trained in a simple environment to reach goals in a small box centred on the agent's initial position. We use Twin-Delayed Deep Deterministic Policy Gradient (TD3) (Fujimoto et al., 2018) with Hindsight Experience Replay (HER) (Andrychowicz et al., 2017), in line with previous work, though any GCRL algorithm could be used. Note that we start the agent at the origin if the provided sub-goals will be relative, otherwise the initial position is randomised. The environment is chosen so that:

1. Its transition function is as close as possible to that of the main environment within the latter's accessible region. For instance, both environments in Figure 1 share the same physics in the corridors, even if not when colliding with walls.

2. Its accessible region is as large as possible in the goal space.

3. The desired goal is in a small (in comparison to the main environment) region of the initial achieved goal (see Figure 1).

We describe this environment as 'simple' since a fully accessible goal space allows easy exploration without trajectories being blocked. Moreover, the desired goal is always nearby compared to goals in the main environment. Such environments are easy to construct by removing obstacles while maintaining physics. In the worst case, it can be formed as a copy of the main environment, except with the desired goal spawning in a small fixed region around the initial achieved goal. Once $\pi_{\mathrm{LL}}$ is trained, its parameters are fixed so that it can be used as a transferable black-box.

## 4.2 Step 2: Trajectory Model Training

We train a Long Short-Term Memory (LSTM) recurrent neural network to predict the full trajectory that $\pi_{\mathrm{LL}}$ will produce, given a goal. This will be used as the first component of our reachability model. To collect data, we run several evaluation episodes of the simple environment with the pre-trained policy $\pi_{\mathrm{LL}}$, collecting full trajectories. We then do several epochs of supervised updates on the LSTM parameters, minimising the Mean Squared Error (MSE) between corresponding goals in the predicted and actual trajectories. Full details of the loss functions can be found in the Appendix.

## 4.3 Step 3: High-Level Planner Training

The planner is trained on the main environment in a round-based fashion, with new landmarks being added to the landmark set progressively. The following two steps are executed in each round:

1. **EXPLORATION**
   For each of several episodes: navigate to the most recently added reachable landmark (according to the reachability model) and then explore randomly (give $\pi_{\mathrm{LL}}$ random subgoals), adding all achieved goals to the landmark's achieved goal buffer (store them as trajectories).

2. **TRAINING**
   For any landmark $l$, with buffer $D_l$, for whom the buffer length $|D_l|$ has reached a fixed threshold:
   - Train the accessibility model using all data from $D_l$.
   - Generate a new landmark $l_{\mathrm{next}}$ from $l$, initialising $D_{l_{\mathrm{next}}}$.

If the initial achieved goal of any episode cannot reach any current landmarks (e.g. at the start, when there are no landmarks), we add the goal itself as a new landmark. Whenever a new landmark is added, it is always marked *unfinished* and we initialise a new buffer for it. The algorithm terminates when all landmarks are marked *finished*. A landmark is marked finished when exploring from it no longer gives novel achieved goals. In the following sections, we provide further details of the components we have presented.

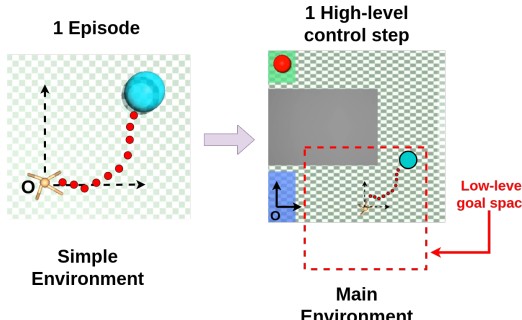

Figure 1: The agent is pre-trained to reach nearby goals in a small environment, free of obstacles and this policy is transferred to the given task environment. The high-level planner gives the low-level policy nearby sub-goals toward the overall task goal.

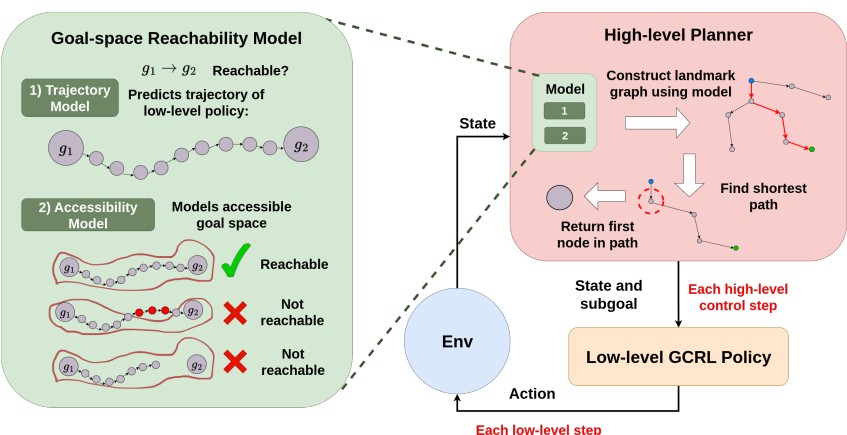

Figure 2: PROMO architecture and inference flow.

## 4.4 LANDMARK GENERATION

We now detail the landmark generation procedure during planner training. To generate a new landmark from a given landmark and a list $D_l$ of achieved goals collected during exploration, we discretise the goal space into a high-dimensional grid of bins (formally hyper-boxes in $G$) and assign a score for each bin. The new landmark is the centroid of the bin with the highest score. We additionally produce a few backup landmarks by treating the score function as a probability distribution and sampling from it.

The score function is a product of three components, enforcing *novelty*, *reachability* and *boundary separation* respectively. We express the score $\mathcal{S}_\mathcal{B}$ for bin $\mathcal{B}$ as:

$$\mathcal{S}_\mathcal{B} := \mathcal{N}_\mathcal{B}\, \mathcal{R}_{\mathcal{B},\, l}\, \mathcal{G}(\mathcal{A}). \tag{1}$$

We describe each term in the product next.

**Novelty Access Score**  $\mathcal{N}_\mathcal{B}$ is a measure of how well points in goal space give access to novel parts of the space, if navigating through them using $\pi_{\text{LL}}$. Let $g_i$ be a goal from $D_l$. We implement $D_l$ so that we have access to the full trajectory of $g_i$. Define $(g_i^1, ..., g_i^{m_i})$ as the remaining segment of $g_i$'s trajectory *after* $g_i$, for some $m_i \in \mathbb{N}$. Then, the novelty access score is the (bin-wise) average proportion of future goals in the trajectory which are novel:

$$\mathcal{N}_{\mathcal{B}} := \frac{1}{n_{\mathcal{B}}} \sum_{\substack{i \\ \text{s.t. } g_i \in \mathcal{B}}} \mathcal{N}(i), \tag{2}$$

$$\text{where} \quad \mathcal{N}(i) := \frac{\sum_{j=1}^{m_i} \mathbb{I}(N_{\pi_{\text{LL}}, l}(g_i^j) = 1)}{m_i}.$$

Here, $n_{\mathcal{B}}$ is the number of goals from $D_l$ lying in bin $\mathcal{B}$ and $\mathcal{N}(i)$ is a novelty access score for the $i$th goal in $D_l$, defined in terms of the indicator function $\mathbb{I}(\cdot)$ and the following function quantifying the novelty of a goal:

$$N_{\pi_{\text{LL}}, L}(g) := \prod_{n=1}^{|L|} (1 - R_{\pi_{\text{LL}}}(l_n, g)). \tag{3}$$

$N_{\pi_{\text{LL}}, L}(g)$ depends on the current set $L$ of landmarks and assumes access to the goal-space reachability function $R_{\pi_{\text{LL}}}(g_1, g_2) : G \times G \to \{0, 1\}$ which we will model in the next section. $N_{\pi_{\text{LL}}, L}(g)$ has value 1 if the input goal is *not* reachable from any current landmark and 0 otherwise.

**Reachability Score**    Next, we present the bin-wise *reachability score*, representing the probability that the centroid of the bin is reachable from the exploration landmark $l$ according to our reachability function:

$$\mathcal{R}_{\mathcal{B}, l} := R_{\pi_{\text{LL}}}(l, c_{\mathcal{B}}). \tag{4}$$

This is needed to make sure the new landmark can actually be reached during planning.

**Boundary Separation Score**    Finally, we introduce a component score which makes sure that landmarks are not generated very close to the edge of the accessible region of the goal space, since this can lead to reachability problems during planning. The score is a Gaussian-smoothed measure of the average accessibility of achieved goals lying within the bin. Consider the bin-wise accessibility score:

$$\mathcal{A}_{\mathcal{B}} := \frac{1}{n_{\mathcal{B}}} \sum_{\substack{i \\ \text{s.t. } g_i \in \mathcal{B}}} A(g_i) \quad , \tag{5}$$

where $A(g)$ is our accessibility function (also modelled in the next section). This measures, on average, whether the goals lying in the bin are accessible. We put this through a Gaussian filter (Haddad et al., 1991) to smooth the boundaries between high and low scores, thereby lowering the score just inside the boundary. Let $\mathcal{G}$ represent the Gaussian filter convolution of $\mathcal{A}_{\mathcal{B}}$ (we use unit variance). This convolution is applied to the accessibility score tensor over bins $\mathcal{B}$. Then the boundary separation score component $\mathcal{G}(\mathcal{A}_{\mathcal{B}})$ is a factor in the final score, defined in Equation 1.

### 4.5 REACHABILITY MODEL

Our reachability model facilitates both graph construction during inference and exploration, as well as reachability/novelty optimisation during landmark generation at training time. It consists of two components: a trajectory model predicting the full trajectory taken by $\pi_{\text{LL}}$ given a goal in the simple environment, and an accessibility model which predicts whether a goal is in the accessible region of the main environment. Given two goals $g_1, g_2 \in G$, we then say $g_2$ is reachable from $g_1$ with respect to $\pi_{\text{LL}}$ if and only if the following two conditions are satisfied:

1. $\pi_{\text{LL}}$'s predicted trajectory successfully reaches $g_2$ in the simple environment
2. The full predicted trajectory is contained in the accessible region of the main environment.

**Suitability**  This is a good approximation since it captures the intuition that an agent needs to 1) be able to travel far enough even without obstructions and 2) have a clear (accessible) path to the goal. While these conditions are usually not enough to *guarantee* reachability in practice, we find that, if used with a correction mechanism (which we describe later), it is enough to achieve excellent results. See Figure 2 for a visualisation of the two conditions. The trajectory model training procedure has been discussed earlier, and all that is left is to train the accessibility model.

**Accessibility Model**  In each round, the accessibility model receives a new batch of goals achieved during exploration and must learn to bound the global distribution of all rounds' data. Any batch-trained, unsupervised density estimator or one-class classifier can be used. The output is a binary value representing inclusion in the learned accessible region. Some example methods are (deep) one-class classifiers (Seliya et al., 2021), deep autoencoders (Zong et al., 2018) and normalising flows (Kobyzev et al., 2020). In our experiments, we use an ensemble of one-class Support Vector Machine (SVM) classifiers (more details in Appendix C), for their ease of implementation, but any method could be substituted. Before high-level training, the model is initially trained on several random exploration episodes. Each subsequent training round naturally fine-tunes the model on data near the chosen exploration landmark.

### 4.6  Planner

At each control step during inference (or exploration), our planner $\pi_{\text{HL}}$ selects a sub-goal for $\pi_{\text{LL}}$ by finding the shortest path through a graph of the current landmarks (plus the current achieved goal and desired goal) using Dijkstra's algorithm and then choosing the first node in the path. Nodes are connected based on reachability. A (unit weight) edge exists between nodes $n_i$ and $n_j$ of the graph if and only if at least one of the following three conditions are met:

$$
\begin{aligned}
&1) \quad i = j \\
&2) \quad n_i \text{ and } n_j \text{ are landmarks and } n_j \text{ was generated from } n_i \\
&3) \quad R_{\pi_{\text{LL}}}(n_i, n_j) = 1 \, .
\end{aligned}
\tag{6}
$$

Due to errors in the reachability model, it is possible (though rare) for $\pi_{\text{HL}}$ not to reach a given node. To mitigate this, we introduce a simple correction mechanism for planning. If a goal is not reached for two consecutive high-level control steps, $\pi_{\text{HL}}$ instead provides a random nearby goal to $\pi_{\text{LL}}$ in the next control step.

### 4.7  Algorithm Termination

The planner training algorithm terminates when all landmarks have been marked *finished*. Each round, we finish the landmark from which exploration was done if the number of novel goals reachable by the would-be next landmark, as a proportion of the round's collected achieved goals, is smaller than a threshold $\epsilon$. We use the value $\epsilon = 0.001$ throughout our experiments. Mathematically, this finishing condition can be expressed as follows, given an exploration landmark $l$, a candidate next landmark $l_{\text{next}}$ and a list $D_l$ of goals achieved during exploration:

$$
\frac{|\{g \in D_l : N_{\pi_{\text{LL}}, L}(g) = R_{\pi_{\text{LL}}}(l_{\text{next}}, g) = 1\}|}{|D_l|} < \epsilon \, ,
\tag{7}
$$

where $N_{\text{LL}, L}$ and $R_{\pi_{\text{LL}}}$ are the novelty and reachability functions presented earlier. If this criterion is satisfied, $l_{\text{next}}$ is discarded and $l$ is marked finished. Otherwise, $l_{\text{next}}$ is added to the set of landmarks. Note that, $l_{\text{next}}$ is only added if a trial episode shows that it is actually reachable from $l$, otherwise the backups are tested for this. If no backups succeed, no landmark is added but $l$ is not finished.

## 5  Experiments

We test our method on challenging new and existing instantiations of the two standard MuJoCo benchmarks for GCRL: AntMaze, a 2D quadruped maze navigation task and Reacher, a 3D robot

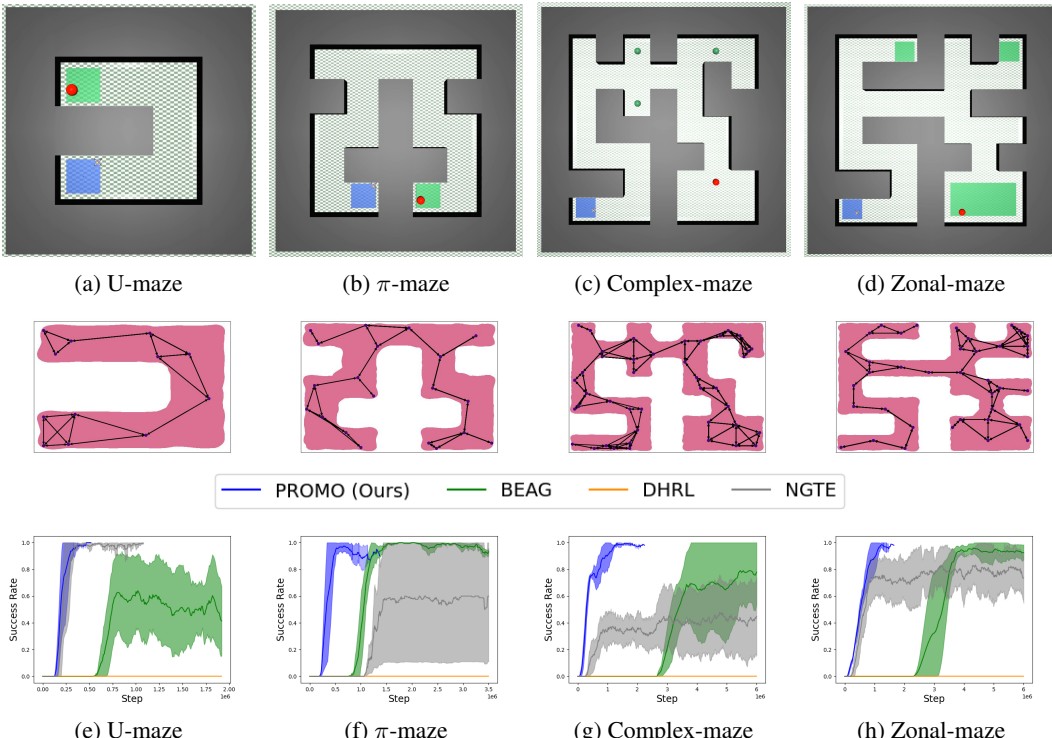

(a) U-maze     (b) $\pi$-maze     (c) Complex-maze     (d) Zonal-maze

(e) U-maze     (f) $\pi$-maze     (g) Complex-maze     (h) Zonal-maze

Figure 3: Visualisations (first row), generated landmark graph with predicted accessible region (second row) and results (third row) for the AntMaze environments, with initial distributions in blue and goal distributions in green. The low-level policy and model were pre-trained for 2.7M total steps (pre-training curves in Appendix E), and then reused for each of the above environments. Our algorithm was trained over 5 seeds, while baselines were trained over 3.

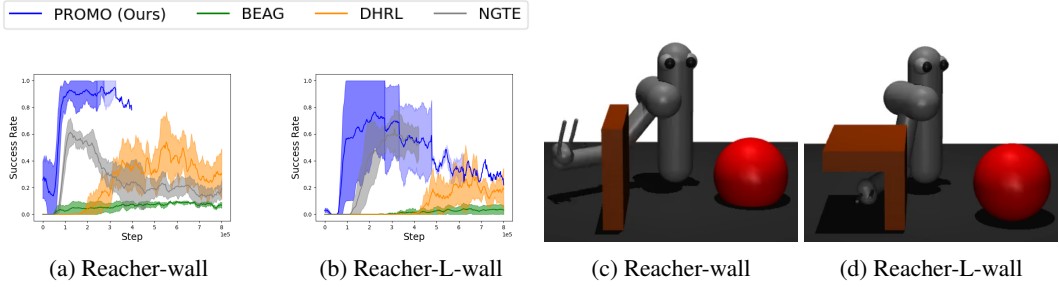

(a) Reacher-wall     (b) Reacher-L-wall     (c) Reacher-wall     (d) Reacher-L-wall

Figure 4: Results for Reacher environments. The low-level policy and model were pre-trained for a total of 1M steps (Appendix E). Our algorithm and all baselines were trained over 5 seeds.

arm spatial goal-reaching task. These are visualised in Figure 3 and full environment details are given in Appendix D. Throughout our experiments, we use sparse reward and the initial state is fixed (with some small local variation). All experiments are evaluated by the average success rate of goal-reaching over 10 evaluation episodes periodically through training, where an episode is considered a success if the goal is reached at any point in its duration. Below, we present comparative results, while detailed ablative studies are given in Appendix F.

## 5.1 COMPARATIVE RESULTS

In this section, we compare our method to the three state-of-the-art graph-based, hierarchical GCRL baselines mentioned earlier: DHRL (Lee et al., 2022), BEAG (Yoon et al., 2024) and NGTE (Park et al., 2024). We first compare the success rates and sample efficiency and then show that merely

transferring a low-level policy does not improve the baselines' results, therefore demonstrating how both levels of our algorithm work *together* to produce excellent results. Finally, we argue that our algorithm produces significantly fewer landmarks and thus explores the space faster.

**Success Curves**   As shown by Figures 3 and 4, our algorithm yields significant improvements not only in sample efficiency, but also in success rates and training stability throughout all experiments. In all environments tested, our algorithm 1) reached its peak performance faster than any other baseline (often by a factor of 3-4x), 2) achieved a near-$100\%$ success rate in all but one environment and 3) achieved a substantially smaller standard deviation between different seeds. It is worth bearing in mind that, unlike the other methods, ours requires a pre-training stage which takes up a noticeable number of steps. However, this need only be performed once and then all environments can enjoy the excellent sample efficiency we have demonstrated. We also conducted a detailed statistical investigation into the performance on Reacher-L-wall (Appendix H). We found that the averaging our algorithm does over non-goal state components does not lead to meaningful increases in error. By examining visualisations of the accessible region, we rather concluded that the errors were due to the choice of the SVM ensemble for the accessibility model. This is not a core part of our methodology and future work could replace the SVM with a more sophisticated neural method.

**Pre-trained Transfer in the Baselines**   To dispel the idea that our method simply achieves good results by virtue of a simple change in the structure of training, we ran the two best performing baselines (BEAG and NGTE) with *our* pre-trained low-level policy, using relative subgoals and frozen parameters, on $\pi$-maze and Zonal-maze. If indeed all our improvements were simply due to pre-training, we would expect the baselines to match our results here. However, both baselines were not able to effectively utilise the policy for either maze, as shown in Figure 11. For example, BEAG only managed to explore half of Zonal-maze. Figure 11d shows a high-level plan generated based on this incomplete exploration.

This failure to adapt is expected, since current hierarchical GCRL methods use end-to-end joint training. This cannot avoid specialising the low-level policy not only to the particular environment, but also to the plans being received from the high-level. It is therefore unsurprising that the baselines cannot take and adapt a generalist policy. By contrast, our method uses the generalist policy in a specific way to model reachability and then uses this model to generate well spaced out landmarks. These results provide strong evidence for the novelty of our transfer-based reachability model.

**Number of Landmarks**   Our method produces significantly fewer and therefore more informative landmarks compared to the previous state of the art. Figure 10 shows some examples of the landmark graphs produced by DHRL, BEAG and NGTE (taken from their papers) alongside those produced by our algorithm. Since exploration is done from the furthest advanced landmark, our method wastes less time closer to the start and expands the covered space faster, leading to better sample efficiency, as demonstrated by the comparative results.

## 5.2   HIGH-DIMENSIONALITY AND NEURAL ACCESSIBILITY MODELS

GCRL methods either focus on high-dimensional visual goal/state spaces for simple, short-horizon tasks (Nasiriany et al., 2019; Eysenbach et al., 2019) or, much more commonly, on low-dimensional goal spaces for complex, long-horizon planning (NGTE, BEAG, DHRL). Our method falls into the latter category. We have therefore used the SVM accessibility model in our experiments to prioritise the ability to model complex topologies over compatibility with very high-dimensions. To use our method in a high-dimensional context, the SVM may need to be replaced by a neural density estimator (thresholded on probability). While testing in higher dimensions is out-of-scope in this paper (as with the SOTA baselines), we sought to prove the concept of a neural accessibility model by substituting a normalising flow (NF) Kobyzev et al. (2020) density estimator, specifically the Augmented Real NVP flow (Huang et al., 2020), and tested on U-maze and Reacher-wall. For Reacher-wall, we used a single NF and for U-maze we used an ensemble.

Figure 5 shows that the NF accessibility model produced good results in these tasks over 5 seeds, though two of the Reacher-wall seeds had to be rerun due to policy collapse caused by underfitting in the NF model. The density estimator expressiveness problem is orthogonal to our algorithm and future work could find a more suitable density estimator or anomaly detection method. Deep SVDD

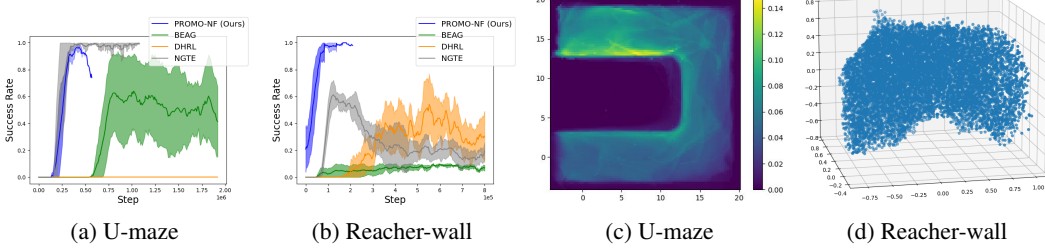

| (a) U-maze | (b) Reacher-wall | (c) U-maze | (d) Reacher-wall |

Figure 5: Left two: results using NF accessibility model for U-maze and Reacher-wall. Right two: visualisations of the resulting accessible regions. U-maze region given by the probability heatmap and Reacher-wall region given by a point cloud.

(Ruff et al., 2018) could be appropriate as it is a neural version of one-class SVM (anomaly detection rather than density estimation). Nevertheless, these results show that our algorithm can, in principle, be used with a purely neural (and therefore high-dimension compatible) reachability model.

### 5.3 COMPUTATION

We ran our experiments on a commodity CPU node with 8 cores and 24GB of memory. We tested the wall clock time on a single machine, demonstrating that the pre-training does not add any extra time per environment step, allowing us to gauge the time saved by pre-training simply by looking at the environment steps in comparison with the baselines. Full details of this experiment are in Appendix I.

## 6 LIMITATIONS AND FUTURE WORK

There are two scope trade-offs we have made in this work: 1) non-goal state averaging and 2) restriction to low-dimensional experiments. Firstly, the trajectory model averages over non-goal state information, which might in theory be slightly rudimentary for some applications. However, our statistical analysis showed that this did not impact the experiments in a meaningful way, suggesting that averaging over non-goal state is indeed applicable to a wide range of complex environments. Nevertheless, we provide a straightforward possible way to incorporate non-goal state into trajectory modelling and landmark generation in Appendix A. Future work could use this to test on non-stationary environments (currently not solved by GCRL), where the effect of non-goal state might be more significant. Finally, future work could test other neural density estimators and ensembling techniques to more accurately model accessibility, expanding the experiments to include high-dimensional goal spaces.

## 7 CONCLUSIONS

Our goal was to examine whether transfer-augmented reachability modelling could provide benefits to GCRL, especially in terms of sample efficiency. Through our experiments, we have answered this question overwhelmingly in the affirmative. Moreover, we even achieved significantly better overall performance and stability than the previous state-of-the-art methods. We hope that this work will stimulate further interest in the broader applications of transfer learning in RL.

## 8 REPRODUCIBILITY STATEMENT

We have included our hyperparameters as well as all parameters for the environments, in order to facilitate reproducibility of our results. We have also provided lists of parameters used for baselines in the Appendix. The important functions for our algorithm's code are given in the supplementary materials. We are currently cleaning our code and, as soon as this is done, the full project will be made available on our GitHub. Finally, our experiments section describes the computational resources used so that other researchers can easily benchmark our algorithm.

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

## A    INCORPORATING NON-GOAL STATE

Here we present a possible method to incorporate non-goal state information into our algorithm, beyond averaging. Currently, we sample the non-goal elements during its (pre-)training. The landmark generation step then uses the reachability model (including the trajectory model) to assess and maximise over certain desirable metrics. This leads to the non-goal state being encoded only in average into the landmark generation. Though this rudimentary encoding is enough to consistently attain better results than existing methods, the results on Reacher-L-Wall show that a more complete encoding might further expand the solvable problem space to tasks where there is high correlation between reachability and all aspects of state. Note that robot arm environments exhibit this property more than navigation tasks since the forearm (whose position is not determined by the achieved goal) can often collide with obstacles.

To encode non-goal state more completely, we need to condition the trajectory model on the full state (subject to zeroing of the achieved goal if using relative coordinates). The next question is how to use this state-conditioned trajectory model during landmark generation. Recall that landmark generation requests the reachability between two goals, effectively averaged over all possible non-goal state. To reproduce this behaviour with a state-conditioned trajectory model, we could randomly sample non-goal state elements and provide them along with the achieved goal input, possibly averaging over multiple samples to improve stability. So far, we have just reproduced the current capabilities of the landmark generator but using a state-conditioned model.

The real power of the state-conditioned model can then be realised during inference to plan paths through the landmark graph which are predicted to be reachable given the actual current state. That is, when the landmark graph is formed at each high-level control step, the edges outgoing from the current achieved goal can be deleted if the destination landmarks are not reachable given the full current state, according to the state-conditioned trajectory model. This method would constitute the initial use of a rough estimate of the full landmark graph (based on state-averaged metrics), followed

by a local refinement based on the actual current state. In principle, this would take issues like forearm collisions into account when planning and should also remove the need for the correction mechanism, as this was only required as a workaround for the non-goal state problem. Future work could implement this procedure and introduce more complex environments where the dynamics of the achieved goal are heavily influenced by all aspects of the state.

## B  TRAJECTORY MODEL TRAINING

Here, we present the loss function for trajectory model training. For relative goals, the trajectory model is a function $F_{\pi_{\mathrm{LL}}} : G' \to \bigtimes_{i=1}^{k} G'$ of one (desired) goal, since the starting goal is always the origin. For absolute goals, the function is $F_{\pi_{\mathrm{LL}}} : G' \times G' \to \bigtimes_{i=1}^{k} G'$ as an additional starting goal input must be provided. We do several episodes of inference in the simple environment and train a Long Short-Term Memory (LSTM) recurrent neural network with $k$ sequential cells to model $F_{\pi_{\mathrm{LL}}}$ by minimising the discrepancy between predicted and actual trajectories. Initial state is randomised, provided the initial achieved goal is the origin if using relative goals. Every cell's input is identical. Given a list $(\tau_1, ..., \tau_N)$ of $N$ length-$k$ achieved goal trajectories, a corresponding list $(g_1, ..., g_N)$ of initial achieved goals and a corresponding list $(h_1, ..., h_N)$ of desired goals, the mean squared error (MSE) loss is:

$$\mathcal{L}_{\mathrm{traj}}^{\mathrm{rel}} := \frac{1}{kN} \sum_{i=1}^{N} \sum_{j=1}^{k} \left\| \tau_i^j - F_{\pi_{\mathrm{LL}}}(h_i)_j \right\|, \tag{8}$$

for relative goals, and

$$\mathcal{L}_{\mathrm{traj}}^{\mathrm{abs}} := \frac{1}{kN} \sum_{i=1}^{N} \sum_{j=1}^{k} \left\| g_i^j - F_{\pi_{\mathrm{LL}}}(g_i, h_i)_j \right\|, \tag{9}$$

for absolute goals. Here, we have used the notation $\tau_i^j$ to mean the $j$th goal in the $i$th trajectory of $(\tau_1, ..., \tau_N)$. Given the success threshold $\delta_r$ and the accessibility function $A(g)$, the reachability function is then expressed as:

$$R_{\pi_{\mathrm{LL}}}(g_1, g_2) = \begin{cases} 1, & \begin{aligned} &\text{if} \quad \left\| F_{\pi_{\mathrm{LL}}}(g_2 - g_1)_k - (g_2 - g_1) \right\| < \delta_{\mathrm{r}} \\ &\text{and} \quad A(F_{\pi_{\mathrm{LL}}}(g_2 - g_1)_i + g_1) = 1 \\ &\forall \, i \in \{1, ..., k\}, \end{aligned} \\ 0, & \text{otherwise} \end{cases} \tag{10}$$

for relative goals, and

$$R_{\pi_{\mathrm{LL}}}(g_1, g_2) = \begin{cases} 1, & \begin{aligned} &\text{if} \quad \left\| F_{\pi_{\mathrm{LL}}}(g_1, g_2)_k - g_2 \right\| < \delta_{\mathrm{r}} \\ &\text{and} \quad A(F_{\pi_{\mathrm{LL}}}(g_1, g_2)_i) = 1 \\ &\forall \, i \in \{1, ..., k\}, \end{aligned} \\ 0, & \text{otherwise} \end{cases} \tag{11}$$

for absolute goals.

## C  ACCESSIBILITY MODEL TRAINING

For the accessibility model, we trained a Support Vector Machine (SVM) classifier which uses a data batch of collected achieved goals to bound a region of the goal space as the accessible region, classifying inclusion in it. Since we only have access to positive data points (achieved goals), we use a *one-class* version of SVM. Additionally, since the data is collected progressively, we train a new SVM each round, on only that round's exploration data, and consider the overall region to be

the union of the regions modelled by the individual components. Therefore, the model is actually an *ensemble* of one-class SVM classifiers. Since each round's data is directly tied to a new base SVM, the initial exploration (before high-level training) is neither possible nor needed with this approach (as it would be for a single global model). We chose this method since it can bound relatively complex data distributions while being easy to implement with `scipy`. However, for high-dimensional goal spaces, a neural method such as normalising flows or neural autoencoders would be a better choice.

Finally, to speed up inference, our implementation stores a bounding box for the data used to train each base SVM. When querying inclusion of a point in the union, the bounding boxes are first checked (a much faster operation) and then only the base SVMs corresponding to the containing bounding boxes are checked. This trick greatly improves inference speed for large goal spaces like AntMaze, but would not be needed for a single, global accessibility model.

## D ENVIRONMENTS

### D.1 ANTMAZE

AntMaze consists of a robot ant quadruped which must navigate a maze to the end goal. All mazes were trained and evaluated with a success threshold of $1$ (in contrast to previous works which evaluate on a more lenient value of $5$). The goal is sampled uniformly from either a small zone, multiple small zones or multiple points. The goal distribution is kept the same for training and evaluation. Our simple environment was set to a 20 x 20 wall-free box centred on the origin, with training success threshold 1. The low-level policy was trained once per seed for 2.5M steps, with the trajectory model trained for a further 200K steps. It was then transferred to the following four mazes:

- U-maze: 24 x 24, 600 steps
- $\pi$-maze: 40 x 40, 1200 steps
- Complex-maze: 56 x 56, 2000 steps
- Zonal-maze: 56 x 56, 2500 steps.

### D.2 REACHER

Whereas AntMaze's goal space is 2-dimensional, we consider two 7-DoF robot arm environments, where the end effector must reach a given goal in 3-dimensional space. The first, Reacher-wall, contains a simple vertical wall to cross over whereas the second, Reacher-L-wall, contains two walls that form an L, where the end effector starts in an area trapped by the walls. In both environments, the initial position is randomised in a small box and the goal distribution is constrained (both during training and evaluation) to a small box on the other side of the obstacle. We use a training success threshold of $0.1$ and an evaluation success threshold of $0.2$. To pre-train our low-level policy, the simple environment has the same dimensions as the main environments but does not contain any walls. Here, the initial position is randomised and the goal is sampled from a small box ($0.5 \times 0.5 \times 0.5$) centred on the initial position. The arm's reach (true accessible region without obstacles) is roughly $0.8$ units either side and about $0.6$ units forward (though it is curved) We trained both the low-level policy and trajectory model for 500K steps each.

## E LOW-LEVEL TRAINING CURVES

We trained the low-level policy for 2.5M steps on our simple AntMaze environment assuming relative goals. We then trained the trajectory model for 200K steps. For Reacher, both the policy and model were trained for 500K steps each. Since the trajectory model training consists of offline data collection, followed by several epochs of updates, we cannot show trajectory model success curve against environment steps. Instead we simply shade in the appropriate number of steps after the policy training curve. The curves for both AntMaze and Reacher are shown in Figure 6, averaged over 5 seeds.

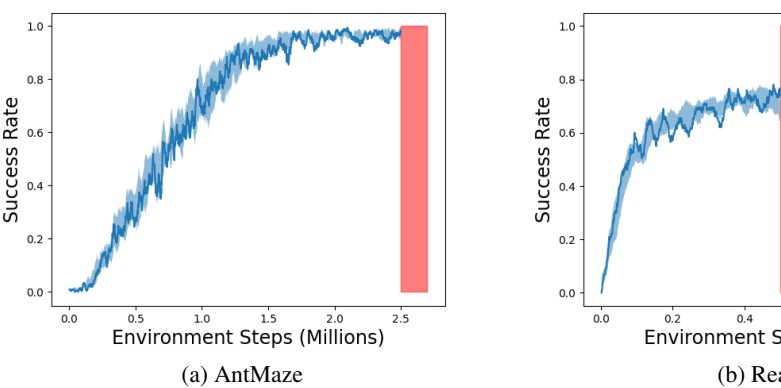

Figure 6: Low-level policy training curves, averaged over 5 seeds and smoothed. Red shaded area shows the additional steps used for trajectory model training.

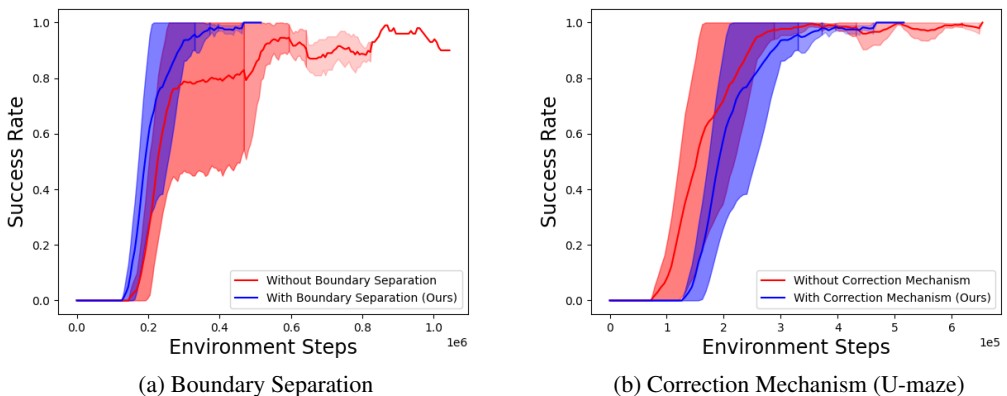

Figure 7: Ablative Results

# F    ABLATIONS

To show robustness and component contribution, we provide the following ablations in AntMaze:

1. Without boundary separation term in landmark generation (U-maze, testing the term's contribution). **Result:** performance, efficiency and stability noticeably degraded. Figure 7a

2. Without correction mechanism (U-maze and $\pi$-maze, testing the mechanism's contribution). **Result:** U-maze shows no significant effect but $\pi$-maze shows some degradation in stability as well as more steps to terminate. Figures 7b and 8a

3. With action noise added to $\pi_{LL}$ in high-level training/evaluation (U-maze, testing robustness to both stochasticity and mismatched high/low-level transition dynamics). **Result:** robust performance and efficiency well maintained, though large action noise values seem to slightly increase the time to terminate. Figure 8b

4. Varying landmark termination threshold $\epsilon$ (U-maze, examining hyperparameter-sensitivity). **Result:** smaller values give good results whereas larger values cause early termination, as expected. Figure 9a

The results are given in Figures 7, 8 and 9.

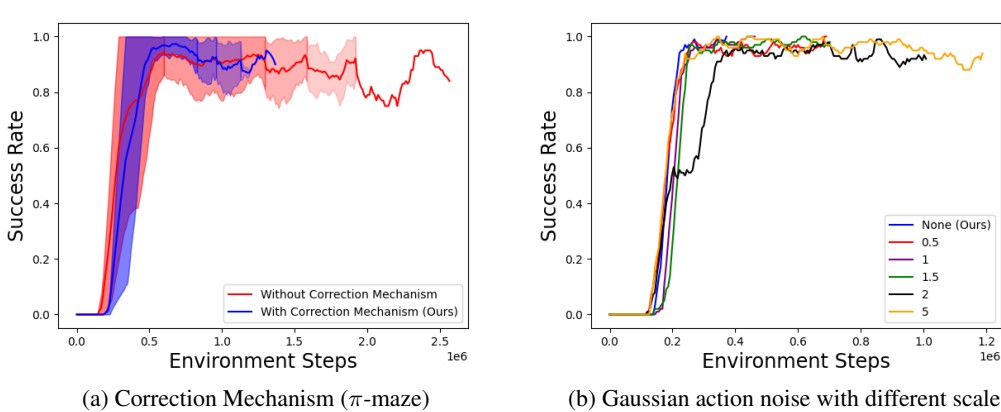

(a) Correction Mechanism ($\pi$-maze)  (b) Gaussian action noise with different scales

Figure 8: Ablative Results

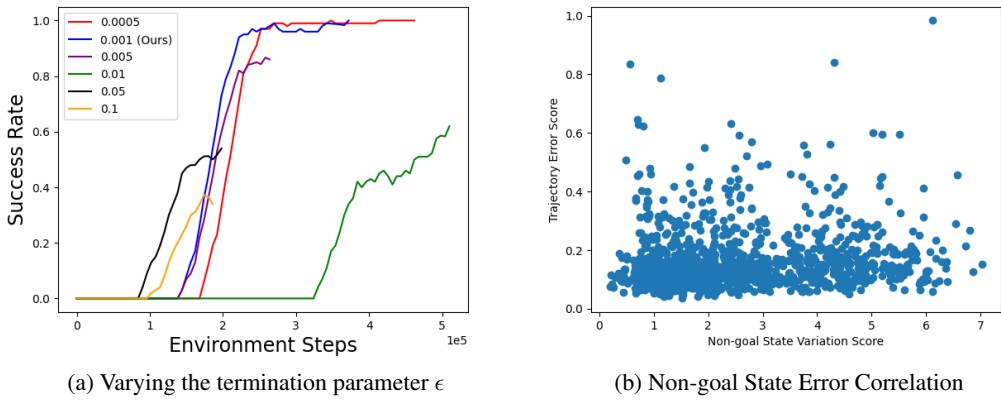

(a) Varying the termination parameter $\epsilon$  (b) Non-goal State Error Correlation

Figure 9: Ablative Results / Statistical Investigation

## G    LANDMARKS

Figure 10 shows visual comparisons of the landmarks generated by BEAG and DHRL (figures presented in their papers) with our landmarks.

## H    REACHER INVESTIGATION

While our algorithm performed the best in both Reacher environments, there were nevertheless some errors preventing full solution, especially for Reacher-L-wall. By inspection of video, we noticed that the arm typically becomes stuck colliding with wall(s) whenever there is a breakdown, suggesting unreachable landmarks and therefore false positives in reachability prediction (which is performed during landmark generation). In order to explain this, we formulated two possible (orthogonal) theories to investigate:

1. **Non-goal state theory:** Since our trajectory model training procedure averages over non-goal components of the state, this information is not encoded with sufficient detail, leading to trajectory prediction inaccuracies and, in turn, false positives in reachability prediction. For example, collisions between the forearm (as opposed to the end effector) and the walls are not taken into account. The correction mechanism is rudimentary and unable to compensate for this when the wall topology is complex.

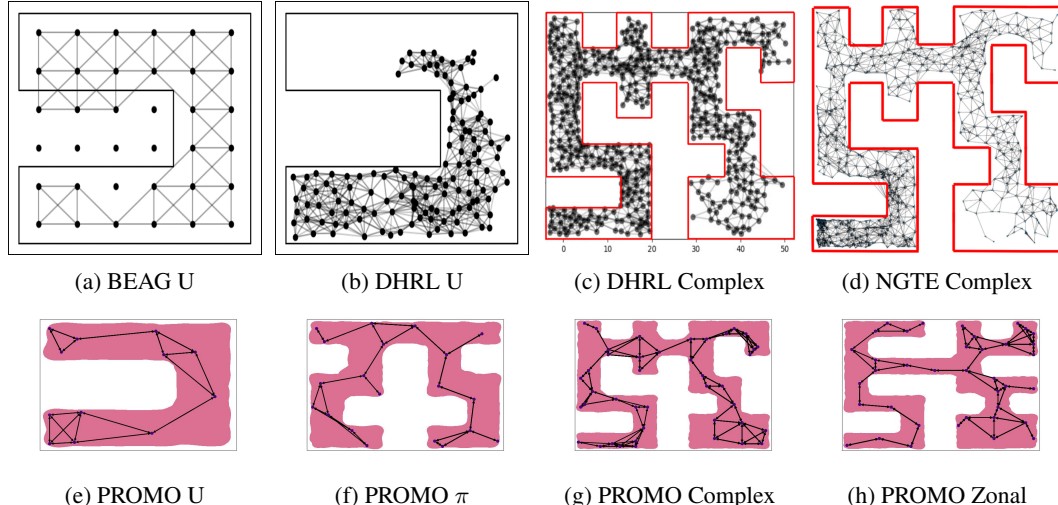

(a) BEAG U  (b) DHRL U  (c) DHRL Complex  (d) NGTE Complex

(e) PROMO U  (f) PROMO $\pi$  (g) PROMO Complex  (h) PROMO Zonal

Figure 10: Various AntMaze environment landmark graphs visualised for DHRL, BEAG, NGTE and PROMO (ours).

2. **Accessibility model theory:** Our one-class SVM ensemble is not flexible enough to encode the non-accessibility resulting from the thin and intricate shapes of obstacles, leading to false positives in accessibility and therefore reachability prediction.

**Non-goal state investigation** For the first theory, we investigate whether there is a positive correlation between variation in initial non-goal state components and average predicted trajectory error, for similar initial achieved goals. Here, we split the achieved goal space into a high-dimensional grid as before and calculate a metric on each bin, given trajectories collected through trajectory model training (in the simple environment). Specifically, if $\{\tau_1^{\text{ac}, \mathcal{B}}, ..., \tau_m^{\text{ac}, \mathcal{B}}\}$ are the $m$ actual full-episode trajectories collected such that their initial states are $\{s_1^{\mathcal{B}}, ..., s_m^{\mathcal{B}}\}$ and their initial achieved goals are all contained in $\mathcal{B}$, we calculate the *non-goal state variation* score of $\mathcal{B}$ as the root mean square deviation of the initial states:

$$V_\mathcal{B} := \sqrt{\frac{1}{m} \sum_{i=1}^m \|s_i^\mathcal{B} - \mu^\mathcal{B}\|^2}, \quad \text{where} \quad \mu^\mathcal{B} := \frac{1}{m} \sum_{i=1}^m s_i^\mathcal{B} . \tag{12}$$

If the bin size is sufficiently small, this should measure the variation of non-goal state while keeping the achieved goal relatively fixed. We obtain the bin size in each dimension by dividing the goal space's total range for that dimension by 10. If no collected trajectories started from $\mathcal{B}$, the score for $\mathcal{B}$ is undefined. Next, we use the trained trajectory model to predict the trajectory $\tau_i^{\text{pr}, \mathcal{B}}$ for each actual trajectory $\tau_i^{\text{ac}, \mathcal{B}}$, given the initial achieved goal and the desired goal for $\tau_i^{\text{ac}, \mathcal{B}}$. We then calculate the *trajectory error* score for $\mathcal{B}$ as:

$$E_\mathcal{B} := \frac{1}{mk} \sum_{i=1}^m \sum_{j=1}^k \|g_{i,j}^{\text{ac}, \mathcal{B}} - g_{i,j}^{\text{pr}, \mathcal{B}}\|, \tag{13}$$

where $k$ is the episode length and $g_{i,j}^{\text{ac}, \mathcal{B}}$ and $g_{i,j}^{\text{pr}, \mathcal{B}}$ are $j$th actual and predicted goals in the $i$th actual and predicted trajectories of $\mathcal{B}$ respectively. Again, this score is undefined for empty bins. Finally, we scatter-plot $E_\mathcal{B}$ against $V_\mathcal{B}$ for each non-empty bin $\mathcal{B}$, expecting to see an increasing pattern if the error is positively correlated with non-goal state variation. In addition, we perform a Spearman's rank correlation test (Spearman, 1904) on the (ordered) lists of error and variation scores to determine whether there is a statistically significant positive correlation. This is an appropriate test as the correlation is not assumed to be linear nor, since all values are positive, Gaussian/symmetric.

We ran $10,000$ evaluation episodes on a trained trajectory model in the Reacher simple environment. The Spearman's test produced a very weak positive correlation of $\rho = 0.1296$, with a one-tailed (testing only positive correlation) $p$-value of $p = 3.664 \times 10^{-6}$. Using the common significance level choice of $\alpha = 0.05$, we see that $p < \alpha$. The correlation is plotted in Figure 9b. We surmise that the correlation, while slight, is statistically significant. In conclusion, while the non-goal state issue contributes very mildly towards reachability modelling errors, it likely does not account for the issues seen in Reacher-L-wall.

**Accessibility model investigation** For the second theory, we first visualise the trained accessible region as a 3D point cloud for both reacher environments. This visualisation indeed shows that, in both cases, the learned region wrongly envelopes the walls (while correctly excluding regions inaccessible due to arm reach). This was confirmed by sampling 10000 goals both within and adjacent to the obstacle and observing that both sets of goals were fully accessible according to the model. The effect of this should be that, when generating landmarks, regions just on the other side of a wall are wrongly seen as reachable, and therefore may be generated as new landmarks.

We then inspected the videos more closely and ascertained that the collisions were happening only with the top (horizontal) wall, not the vertical wall. This leads us to the following conclusion about the unreachable landmarks. Since the angles of the forearm and elbow make it in any case difficult to access the region just on the other side of the vertical wall, and since this is not true for the horizontal wall, the spurious landmarks are only generated close-by on the other side of the horizontal wall. This explains the failures in Reacher-L-wall as well as the success of Reacher-wall, which does not have a horizontal wall. The issue is therefore that very thin, intricate obstacles may push the limits of the accessibility model's expressiveness. We note, however, that there are some seeds in Reacher-L-wall which generate near-$100\%$ results and, even in the bad seeds, the success rate is still higher than the best baseline. We attribute this to the highly informative nature of other landmarks nearby to spurious ones.

## I WALL CLOCK TIME

In order to demonstrate that the computational overhead of pre-training is not disproportionate to high-level training, we performed timed runs for all three algorithm stages with U-maze on a single machine, a Dell Intel I7 laptop with $16.5$GB of memory. This was chosen since our cluster produces very unpredictable speeds. The times were then divided by the total number of environment steps for the corresponding stage and the time-per-step values were compared. In particular, policy pre-training took $37,271,471$ ms ($\approx 10.4$ hrs) for $2,500,000$ steps, trajectory model training took $1,768,500$ ms ($\approx 29.5$ min) for $200,000$ steps and the planner took $12,145,708$ ms ($\approx 3.4$ hrs) for $739,349$ steps. The time-per-step rates are therefore $15$, $9$ and $16$ milliseconds per step respectively. This shows that there is no increased computational burden for pre-training, up to the number of environment steps used.

We were unable to perform fair wall clock time tests on the baselines since they required more memory than was available on the laptop. However, since the pre-training time-per-step rates are similar to or lower than planner training, we can use the number of environment steps as a proxy to compare overall computational burdens (at least to rule out added overhead based on our pre-training contribution). As shown in our comparative results, our method significantly improves sample efficiency if given a one-time pre-training budget. For the price of 2.7M pre-training steps, our algorithm saves around 3M steps in both Complex-maze and Zonal-maze and attains significantly better success rates in Reacher, thus justifying the trade-off.

## J TRANSFER IN THE BASELINES (INVESTIGATION)

To prove that our comparative results cannot be matched simply by incorporating transfer into existing methods, we tried to transfer our pre-trained policy to BEAG and NGTE. However, this returned poor results, shown in Figure 11.

## K PSEUDOCODE AND HYPERPARAMETERS

---

**Algorithm 1** PROMO Planner Training

---

**Require:** Steps per round $n_{\text{steps}}$, pre-trained low-level policy $\pi_{\text{LL}}$, maximum control step interval $k$, initial state distribution $\rho_S$, low-level goal space $G_{\text{LL}}$, environment `env`, state-goal mapping $\phi$, train threshold $n_{\text{tr}}$

1: **procedure** TRAINPLANNER
2:     Train accessibility model on initial exploration data (explore directly from initial states)
3:     Landmark set $L \leftarrow \{\}$
4:     Buffers $\leftarrow$ `List()`
5:     **for** $r = 1, 2, \ldots$ **do**
6:         Buffers $=$ `Explore(`$L$`, Buffers)`
7:         **for** $l \in L$ **do**
8:             **if** $|D_l| \geq n_{\text{tr}}$ **then**
9:                 Train Accessibility Model using $D_{l_r}$
10:                 Calculate $\mathcal{S}_\mathcal{B}$ over grid bins $\mathcal{B}$
11:                 $l_{\max} \leftarrow \arg\max_\mathcal{B} \ \mathcal{S}_\mathcal{B}$
12:                 Check finishing condition for $l$ using $l_{\max}$              ▷ Using 7
13:                 **if** Finishing condition satisfied **then**
14:                     Discard $l_{\max}$
15:                     Mark $l$ **finished**
16:                 **end if**
17:                 Sample $q$ backups $b_1, \ldots, b_q$ as bin centroids     ▷ With bin $\mathcal{B}$ probability $\frac{\mathcal{S}_\mathcal{B}}{\sum_\mathcal{B} \mathcal{S}_\mathcal{B}}$
18:                 **for** Candidate $l_{\text{next}}$ in $(l_{\max}, b_1, \ldots, b_q)$ **do**
19:                     Trial episode to verify reachability of $l$ from $l_r$
20:                     **if** verified **then**
21:                         Add $l_{\text{next}}$ to $L$
22:                         Initialise $D_{l_{\text{next}}}$
23:                         break
24:                     **else**
25:                         Discard $l_{\text{next}}$
26:                     **end if**
27:                 **end for**
28:             **end if**
29:         **end for**
30:         **if** All landmarks **finished then**
31:             break
32:         **end if**
33:     **end for**
34:     **return** $L$
35: **end procedure**

---

Table 1: **PROMO Hyperparameters (AntMaze)**

|  | Final Value | Range Tried |
|---|---|---|
| $n_{\text{steps}}$ (expl. steps per episode) | 10000 | 2000 - 16000 |
| $n_{\text{bins}}$ (landmark gen grid) | 50 | 50 |
| $\epsilon$ (finishing condition) | 0.001 | 0.001, 0.005 |
| SVM RBF kernel $\nu$ | 0.01 | 0.01 - 0.1 |
| SVM RBF kernel $\gamma$ | 0.1 | 0.01 - 0.1 |
| LSTM hidden dim | 64 | 64, 128, 256 |
| LSTM batch size | 500 | 200, 500, 1000 |
| LSTM num train epochs | 3000 | 1000, 3000, 10000 |
| LSTM train steps | 200K | 200K |
| $\pi_{\text{LL}}$ Gaussian action noise $\sigma$ | 2. | 0.1 - 2 |
| $\pi_{\text{LL}}$ learning rate | 0.001 | 0.001 |
| $\pi_{\text{LL}}$ batch size | 256 | 256 |
| $\pi_{\text{LL}}$ learning rate | 0.001 | 0.001 |
| $\pi_{\text{LL}}$ train steps | 2.5M | 2.5M, 3M |

---

**Algorithm 2** Explore (Subroutine)

---

**Require:** Steps per round $n_{\text{steps}}$, planner $\pi_{\text{HL}}$, pre-trained low-level policy $\pi_{\text{LL}}$, maximum control step interval $k$, initial state distribution $\rho_S$, low-level goal space $G_{\text{LL}}$, environment `env`, state-goal mapping $\phi$

1: **procedure** EXPLORE($L$, $\{D_l\}_l$)
2:     **Parameters:** Landmark set $L$, landmark buffers $\{D_l\}_l$
3:     step $\leftarrow 0$
4:     **while** step $< n_{\text{steps}}$ **do**
5:         $s \sim \rho_S$
6:         $l \leftarrow$ most recent reachable landmark or $\phi(s)$
7:         Append $\phi(s)$ to $D_l$
8:         **while** episode not terminated **do**
9:             **if** control step **then**
10:                 **if** reached $l$ this episode **then**
11:                     Subgoal $h \sim$ `Uniform(`$G_{\text{LL}}$`)`
12:                 **else**
13:                     Subgoal $h \leftarrow$ `Plan(`$\phi(s)$`, `$l_e$`, `$L$`)`
14:                 **end if**
15:             **end if**
16:             Low-level action $a \leftarrow \pi_{\text{LL}}(s, h)$
17:             Next state $s \leftarrow$ `env.step(`$a$`)`
18:             Add $\phi(s)$ to $D_l$
19:         **end while**
20:     **end while**
21:     **return** buffers $\{D_l\}_l$
22: **end procedure**

---

**Algorithm 3** Plan (Subroutine)

---

**Require:** Reachability function $R_{\pi_{\text{LL}}}$

1: **procedure** PLAN($g$, $h$, $L$)
2:     **Parameters:** Achieved goal $g$, desired goal $h$, landmark set $L$
3:     Construct landmark graph
4:     Add $g$, $h$ to graph using $R_{\pi_{\text{LL}}}$
5:     Shortest path $(h_1, ..., h_p, h) \leftarrow$ `Dijkstra(`$h$`)`
6:     **return** $h_1$
7: **end procedure**

---

Table 2: **PROMO Hyperparameters (Reacher)**

| | **Final Value** | **Range Tried** |
|---|---|---|
| $n_{\text{steps}}$ (expl. steps per episode) | 8000 | 2000 - 20000 |
| $n_{\text{bins}}$ (landmark gen grid) | 50 | 50 |
| $\epsilon$ (finishing condition) | 0.001 | 0.001, 0.005 |
| SVM RBF kernel $\nu$ | 0.01 | 0.01 - 0.1 |
| SVM RBF kernel $\gamma$ | 0.1 | 0.01 - 0.1 |
| LSTM hidden dim | 64 | 64 |
| LSTM batch size | 100 | 100, 100, 500 |
| LSTM num train epochs | 500 | 200, 500 |
| LSTM train steps | 500K | 500K, 1M |
| $\pi_{\text{LL}}$ Gaussian action noise $\sigma$ | 2. | 0.1 - 2 |
| $\pi_{\text{LL}}$ learning rate | 0.001 | 0.001 |
| $\pi_{\text{LL}}$ batch size | 256 | 256 |
| $\pi_{\text{LL}}$ train steps | 500K | 500K |

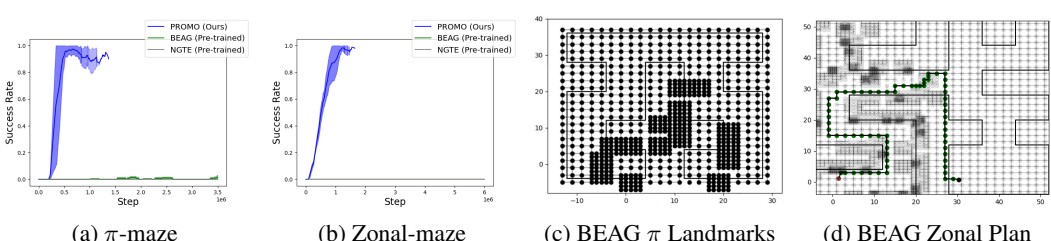

(a) $\pi$-maze      (b) Zonal-maze      (c) BEAG $\pi$ Landmarks      (d) BEAG Zonal Plan

Figure 11: Left two: baselines cannot adapt our pretrained (fixed) low-level policy to AntMaze tasks. Right two: visualisations of graphs generated by the BEAG baseline with our pre-trained low-level policy. Note the generated plan in Zonal-maze goes through the walls since only half of the maze has been explored.

Table 3: **Hyperparameters for DHRL and BEAG.** We experimented with various hyperparameter configurations, as well as different numbers of landmarks.

|  | DHRL | BEAG |
|---|---|---|
| initial episodes without graph planning | 75 | - |
| gradual penalty | 1.5-5.0 | - |
| high-level train freq | 10 | - |
| failure count threshold $\tau_n$ | - | 100,200,300,400 |
| failure condition threshold $\tau_t$ | - | 1,3,5,7 |
| number of landmarks | 300-600 | 36-256 |
| hidden layer | (256, 256) | (256, 256) |
| actor lr | 0.0001 | 0.0001 |
| critic lr | 0.001 | 0.001 |
| target network soft update rate $\tau$ | 0.005 | 0.005 |
| discount factor $\gamma$ | 0.99 | 0.99 |
| batch size | 1024 | 1024 |
| target update freq | 10 | 10 |
| actor update freq | 2 | 2 |

