# OpenReview forum: "Policy Transfer for Improved Sample Efficiency in Goal-Conditioned Reinforcement Learning"
_ICLR.cc/2026/Conference — ICLR 2026 Conference Withdrawn Submission_

### Official Review · Reviewer_JLZB · 2025-10-29

**Soundness:** 3
**Presentation:** 3
**Contribution:** 2
**Rating:** 2
**Confidence:** 5

**Summary:**

This paper proposes PROMO (Progressive Reachability Optimisation MOdelling), a hierarchical framework for Goal-Conditioned Reinforcement Learning (GCRL) that aims to improve sample efficiency by reusing a pre-trained, transferable low-level policy across similar environments. PROMO further integrates a trajectory model to predict the motion of the low-level policy and an accessibility model to estimate valid goal regions, which are jointly used for landmark generation and graph expansion. While the method largely relies on engineering-driven design choices (e.g., TD3+HER, LSTM trajectory prediction, one-class SVMs), the empirical results demonstrate that PROMO can enhance sample efficiency and stability in GCRL benchmarks.

**Strengths:**

1. Clear and modular problem formulation:
PROMO offers a clean separation between low-level skill learning and high-level planning in the GCRL setting.
Unlike prior methods (e.g., DHRL, BEAG, NGTE) that jointly train both levels, PROMO formalizes the transfer of a pre-trained low-level controller as a fixed reusable module. This modularization improves conceptual clarity and provides a practical recipe for reusing skills across related environments.

2. Model-based reachability formulation:
The paper introduces an explicit reachability model R(g1, g2), defined as the conjunction of a learned trajectory predictor and an accessibility estimator. While simple, this definition represents a step toward formalizing how graph-based hierarchical policies can rely on learned dynamics priors instead of raw distance or value metrics. This shift from distance-based heuristics to a model-based reachability function adds conceptual novelty and a foundation for future model-based planning extensions.

3. Improved sample efficiency through skill transfer:
The empirical results convincingly show that reusing a pre-trained goal-reaching skill leads to faster convergence and greater stability (3~4× sample efficiency gains) across AntMaze and Reacher variants. This provides the first systematic demonstration that transferable skills can meaningfully accelerate hierarchical GCRL without joint retraining, supporting the broader argument that policy reuse can scale beyond single-task settings.

**Weaknesses:**

1. Limited conceptual originality  :
PROMO relies on well-established components: a low-level policy based on TD3+HER, an LSTM-based trajectory predictor, a one-class SVM for accessibility estimation, and a graph-based planner. While this combination is practically reasonable, it does not present a fundamentally new academic contribution. PROMO can be viewed more as a system-level integration and structural organization of existing GCRL techniques rather than a conceptually novel framework.

2. Limited transferability:
Although the paper emphasizes a transferable low-level policy, all experiments are conducted within the same domain, using the same robot morphology and dynamics (AntMaze variants). For instance, if the agent encounters a new bottleneck structure such as AntMazeBottleneck, the pre-trained skill could not generalize effectively. Similarly, for robot manipulation tasks involving contact-rich interactions rather than simple goal-reaching, the policy transferability would likely break down.

3. Heuristic dependence in landmark generation:
The landmark scoring function is a simple multiplicative combination of Novelty × Reachability × Boundary Separation.
There is no clear justification for why this multiplicative relationship is optimal, nor is there discussion about the potential need for weighting between the terms. Moreover, the overall exploration behavior appears quite similar to NGTE and BEAG, suggesting that the underlying novelty-driven expansion mechanism has not been substantially advanced.

4. Limited experimental diversity and scalability:
All PROMO experiments are conducted in low-dimensional goal spaces (2D AntMaze, 3D Reacher).
The goal space is also handcrafted, defined directly in terms of coordinates such as (x, y).
This design does not scale to high-dimensional or visual goal spaces, nor to contact-rich manipulation settings.
Recent studies have already begun to use latent goal spaces for more complex, visually grounded tasks in environments such as Robosuite, LIBERO, CALVIN, ManySkill, Kitchen (D4RL), and OGBench (Locomotion & Manipulation tasks)

**Questions:**

Please address the points mentioned in the Weaknesses section above.

In addition, I have a specific concern regarding the claimed sample efficiency and transferability.
You mention that the low-level policy in the AntMaze domain was trained for 2.5M steps separately.
From a single-environment perspective, this makes PROMO appear less sample-efficient than NGTE or BEAG.

The main argument is that PROMO benefits from reusing a fixed pre-trained low-level policy across environments.
However, if we assume that the low-level policy is based on relative subgoals, then NGTE and BEAG could equally reuse the same low-level policy trained in U-maze and keep it fixed when training on Complex-maze.
In that case, the transferability claim of PROMO may not provide a unique advantage.

Would it be possible to evaluate PROMO, NGTE, and BEAG under the same setting where all methods reuse the same pre-trained low-level policy (frozen during high-level training)?
This would allow for a fair comparison focusing solely on high-level graph construction and expansion efficiency, which seems to be the true distinguishing factor of PROMO.

---

> ### Author Response · Authors · 2025-11-28
>
> Thank you for your insightful comments, based on which we have re-evaluated and clarified our contribution. We address your concerns point-by-point:
>
> 1. The trajectory and accessibility models are not “existing GCRL techniques”. Existing GCRL methods do not learn reachability from data at all (as you mentioned in the strengths), let alone with the specific trajectory + accessibility + transfer architecture we have introduced. Learning reachability is hard - we initially tried with just a monolithic MLP but this was prohibitively inefficient due to the ‘quadratic’ data collection complexity (i.e. need to collect enough start goals and from each, enough end goals). We therefore broke the reachability task down into trajectory + accessibility so that, in the main environment, it is only a ‘linear’ data collection problem (accessibility model data) and the quadratic problem is restricted to a much simpler, short-horizon environment. This constitutes a novel insight and execution to serve a novel purpose, which is applicable to all GCRL tasks and is therefore a research contribution rather than a system-level integration for a specific application. Of course, every research contribution will use tools presented in earlier papers (e.g. neural networks), but we have presented a new architecture and the new paradigm of learning reachability, which could open up a completely new avenue for GCRL research. Our two main contributions, as clarified in the general comment and the new related work section, are 1) the first successful low-level policy transfer in GCRL and 2) the first learned reachability model to facilitate landmark generation.
>
> 2. The purpose of transfer in our method is not to go across domains (doing this for low-level policies is impossible since the state/action spaces will mean fundamentally different things). Rather, it is to transfer a policy trained in a simple, short-horizon task (goals close to initial state) to a much more complex, long-horizon task which could not be solved by existing RL. In the AntMazeBottleneck example, this would correspond to incorporating a bottleneck in the low-level environment and then transferring to a maze of any size with arbitrarily many bottlenecks placed at any position. This is a significant application of transfer. Finally note that, based on the results, the SOTA baselines would not match our performance or efficiency in the large mazes even if we were to pre-train a fresh low-level policy before each maze (which would never need to happen anyway).
>
> 3. Reachability and boundary separation are binary terms while the novelty access term is continuous, so there is no need for weighting. This product intuitively means “maximise novelty access, such that reachability and a certain (fixed) boundary margin are preserved”. Moreover, the landmark expansion behaviour is fundamentally different from (and superior to) NGTE, BEAG and DHRL, as can be seen by the fact that ours produces an order of magnitude fewer landmarks while covering the same full goal space and vastly improving sample efficiency. The landmark comparison visualisations have been added to Appendix G of the new revision.
>
> 4. As clarified in section 5.2 of the new revision, most GCRL methods (including NGTE/BEAG/DHRL) tackle complex, long-horizon tasks in low-dimensional goal spaces, while the state dimension can be much higher (31D in AntMaze). Some past methods (e.g. LEAP, RIS) have tackled high-dimensional visual goal spaces but at the cost of task complexity in non-visual environments. For example, LEAP only tested on U-maze, as opposed to larger / more complex mazes. RIS tested on slightly more complex mazes (still much smaller than 56x56) but assumed a uniform initial state distribution, a significant simplifier. To our knowledge, no GCRL method tackles complex, long-horizon tasks of the hardness of our experiments while also testing on visual / high-dimensional tasks. Nevertheless, we have shown in section 5.2 that a neural accessibility model works with our method. With both the accessibility and trajectory models neural, there is nothing stopping our method from working in high dimensions. As for contact rich robotics tasks, these are perfectly fine as long as they are goal-reaching tasks. For example, Pusher is a contact-rich goal-reaching task similar to Reacher. If our method were applied to such a task, the low-level policy would learn to move an object a small distance and the overall policy could then move the object through long, complex trajectories, analogous to how this was done in Reacher.
>
> **CONTINUED NEXT...**

---

> > ### Author Response · Authors · 2025-11-28
> >
> > **...CONTINUED**
> >
> > 5. Even from a single environment perspective, the baseline results do not match our performance or efficiency on Complex-maze, Zonal-maze or either of the Reacher tasks. But even this seriously undersells the transfer benefits of our method.
> >
> > 6. We have now tried to transfer our fixed pre-trained policy with relative sub-goals to NGTE and BEAG. Both of them were unable to utilise the policy, which is unsurprising given that they rely on joint specialisation. See the general comment and section 5.1 of the new revision for details. These results show that our excellent results were due to the combination of a transferred policy and the novel reachability model.

---

### Official Review · Reviewer_vYoq · 2025-10-30

**Soundness:** 2
**Presentation:** 2
**Contribution:** 1
**Rating:** 2
**Confidence:** 4

**Summary:**

The paper introduces a novel algorithm for hierarchical goal-conditioned reinforcement learning, PROMO, where they show that a low-level policy can transfer across many environments in a hierarchical RL setting. PROMO first trains a low-level goal-reaching policy in a simple, obstacle-free setting using TD3 + HER, which is then frozen and reused. A trajectory model (LSTM) predicts how this low-level policy behaves given a goal, and an accessibility model estimates which goals are reachable in the main environment. A high-level planner builds a graph of landmarks (subgoals) that progressively covers the goal space using a reachability scoring function. Planning between landmarks is performed with Dijkstra’s algorithm.

Their method improves sample efficiency compared to stare-of-the-art methods.

**Strengths:**

- their algorithm has good performance on the environments they studied
- their algorithm learned what seem like sensible landmarks

**Weaknesses:**

The paper’s claims and methodology need clearer quantification and exposition. It’s not evident how the authors measure that their landmarks are “fewer” or “more informative,” nor how they determine that the goal space is “covered using only as many landmarks as needed,” since no optimal baseline or comparison is shown. The “near 100% success rate” is also overstated given that the low-level policy is pretrained and transferred rather than zero-shot. The criterion for marking a landmark as “finished” appears arbitrary, lacking clear time or iteration constraints. Figure 2 and Section 4 are confusing, with unclear boundaries between training and inference, and the procedural flow of PROMO is hard to follow without a single unifying formal algorithm.

**Questions:**

- "much fewer, more informative landmarks": how do you quantify that they're "more informative"? Looking at Figure 3, (c-d), it not at all obvious that the landmarks displayed there is the fewest "most informative landmarks." That is assuming that c-d are showing landmarks. It's not clear what "model predictions" is.
- "our method achieves near 100% success rate in almost all environments" - this is expected because you're training/finetuning/transferring. Unless you are saying that you get 100% zero-shot transfer success rate, which I don't believe you are. You pretrain your lower-level policy to reach small goals in a small, obstacle-free environment. It's unsurprising that this is a good transfer policy for complex AntMaze environments.
- you say the landmark generation method "smartly covers the goal space using only as many landmarks as are needed". How do you quantify that only as many as needed are used. Do you know the true "optimal" amount of landmarks and that your method gets no more than that. Where do you show and compare against that?
- "A landmark is marked finished when exploring from it no longer gives novel achieved goals" - what time constraints do you use for this? This seems like an arbitrary criterion.
- Figure 2 is not easy to read. How do I read (2) of the left side. How are the boundaries selected? Are you describing training/action-selection? This is very confusing.
- Section 4 was not easy to read at all. The delineation between training/action-selection was not clear throughout. This section would benefit from a formal algorithm to describe everything together.
- you say your method reaches near 100% success rate in almost all environments, but figure 4b shows that your method is far from 100% if you account for the quite large variance it has. this is seems like an(other) example of overclaiming.

---

> ### Author Response · Authors · 2025-11-28
>
> Thank you for your comments. We address your concerns point-by-point:
>
> 1. Indeed, the claim of the “fewest” or “most informative” set of landmarks was unwarranted and we have now removed these superlatives. Instead, we only claim that they are much fewer and more informative (since they still cover the whole goal space) as compared to the existing methods. We have added side-by-side visualisations of generated landmark graphs for our method vs the SOTA baselines in Appendix G of the new revision. From those, you can see that ours produces an order of magnitude fewer landmarks while still covering the full goal space (modelled goal space is in pink). The hugely improved sample efficiency shows the benefit of this landmark generation improvement. We have also replaced “model predictions” with “generated landmark graph with predicted accessible region” for clarity.
>
> 2. Achieving near-100% success rate is not at all expected. The low-level policy can only move a small distance in free space. Complex-maze and Zonal-maze are extremely large (56x56) with very complicated topologies. This requires a sophisticated landmark generation procedure which can design very long paths through accessible goal space. We have now also tried to transfer our pre-trained policy to BEAG and NGTE, which both failed to use it effectively. This is unsurprising, since they rely on joint training which specialises the low-level policy to the particular task and high-level plans. See the general comment and section 5.1 + Appendix J of the new revision for details. These results demonstrate that our excellent results are not just due to transfer, but the combination of low-level transfer with our high-level reachability model. Note also that none of the SOTA baselines are able to match our success rates.
>
> 3. As mentioned in the first point, we have replaced the superlatives with comparative statements, which can be verified by the visualisations and the improved sample efficiency. We do not claim any theoretical optimality on the number of landmarks.
>
> 4. The whole point of exploring from a landmark is to find novel nearby points. If the surrounding space is no longer novel (because there are now other nearby landmarks), we must stop exploring from it since there are no more useful landmarks we can generate from this landmark. This is the natural criterion for finishing a landmark and is measured precisely in equation 7. Time has no relevance here at all - it is down to the induced (non-)novelty of the surrounding regions based on the current landmark set. Throughout all our experiments, we find that this termination condition works exceptionally well. Note how all graphs terminate shortly after reaching peak performance (except for a degenerate seed in Reacher-L-wall). This even holds true for the new neural accessibility model results. Existing methods do not self-terminate and this is therefore another benefit of our algorithm.
>
> 5. The main flow in Figure 2 describes the inference flow (action-selection). This was already mentioned in the caption, but we have now clarified this in the text referencing the figure. The “Reachability Model” sub-box describes how the reachability model decides whether $g_2$ is reachable from $g_1$. The reachability model is invoked at inference time as well as at training time (both during exploration and landmark generation). 2) of the sub-box describes how the modelled trajectory is compared against the current estimate of the accessible region (modelled by the accessibility model) to ascertain reachability. The reachability model returns True only if the trajectory reaches $g_2$ irrespective of the accessible region AND all points in the trajectory stay within the accessible region, as shown by the three examples. The “boundaries” you are referring to are the estimate of the accessible region (working estimate if at training time), given by the accessibility model. Full details are in the Reachability Model section of the methodology.
>
> 6. We have now added clarifying text throughout section 4 (see the blue highlights) making it clear what is training and what is inference. If you require any more clarification, please let us know and we will be happy to add it. The full pseudocode was already provided in the appendix but we have now referenced it in the method overview. To summarise, sections 4.1-4.3 describe the 3-step training procedure, including landmark generation, which is detailed in section 4.4. Then in sections 4.5-4.7, we simply detail the remaining components we already mentioned. The overall inference flow is visualised in Figure 2 and the planner action-selection is detailed in section 4.6.
>
> 7. The claim is near-100% success rate in almost all environments, meaning five out of the six tasks. This claim is clearly demonstrated by the graphs. Figure 4b is the one out of six that did not achieve this.

---

### Official Review · Reviewer_WPyx · 2025-11-01

**Soundness:** 2
**Presentation:** 2
**Contribution:** 2
**Rating:** 4
**Confidence:** 5

**Summary:**

This paper investigates policy transfer in goal-conditioned reinforcement learning (GCRL) through a hierarchical framework that separates a transferable low-level controller from a high-level planner. The proposed method, named PROMO, aims to improve sample efficiency by reusing a pre-trained low-level policy across environments. Experiments on AntMaze and Reacher tasks demonstrate significant sample-efficiency gains and higher success rates compared to existing hierarchical GCRL baselines.

**Strengths:**

1. The paper addresses an important and still open problem in reinforcement learning: improving sample efficiency in long-horizon sparse-reward goal-conditioned tasks. The research question is interesting and relevant to the field.

2. The empirical section is clear, and the performance improvements over baselines such as DHRL, BEAG, and NGTE are well presented.

3. The paper provides detailed descriptions of the algorithmic components and training pipeline, which helps reproducibility.

**Weaknesses:**

***1. Limitation of novelty:*** The main idea of separating high- and low-level policies is not new in hierarchical reinforcement learning. The structure closely follows prior work such as HIRO, HAC, and other two-level GCRL frameworks. Although the authors introduce transfer of the low-level policy, the proposed design does not clearly resolve the fundamental bottlenecks known in HRL systems. In particular, the high-level policy still relies on an abstract transition model without sufficient domain knowledge or grounded dynamics understanding. It is unclear how the high-level planner can reason about long-term transitions or choose sub-goals efficiently when the reachability model and trajectory predictor are both learned in limited or simplified environments. The paper would benefit from a deeper analysis or theoretical argument explaining why this decomposition can generalize beyond the pre-training domain.

***2. The design of LSTM.*** Based on Weakness1, it is more questionable about the effectiveness of the LSTM trajectory model. Using an LSTM to predict the trajectory of the low-level policy is an acceptable but limited choice. It works reasonably well in simple settings like AntMaze, where dynamics are low-dimensional and repetitive. However, such models struggle to represent multi-modal trajectories or long-range dependencies in complex domains. The experiments indeed demonstrate that while the approach performs well in the maze tasks, its advantage diminishes in the Reacher environment. This observation suggests that the trajectory model is not robust to more complex or highly nonlinear transitions. The authors should discuss how this component scales to more realistic or high-dimensional environments.

***3. Weak baselines and missing analysis.*** The selection of baselines is not sufficiently comprehensive. While the comparisons with DHRL, BEAG, and NGTE are standard, other relevant methods such as SoRB should have been included. In particular, SoRB directly addresses goal-space planning using learned value functions. CO-PILOT directly use the agent's exploration knowledge for planning. Moreover, the paper does not report the training cost or computational overhead of the additional LSTM trajectory model. Given that the method involves multiple training stages (pre-training, trajectory modelling, and high-level planning), reporting wall-clock time or FLOP estimates would strengthen the claim of improved sample efficiency.

**Questions:**

The methodological contribution appears close to that of existing HRL frameworks and even overlaps conceptually with recent transfer-based GCRL approaches. For instance, the ideas in “Policy Transfer for Improved Sample Efficiency in Goal-Conditioned Reinforcement Learning” resemble those discussed in https://arxiv.org/abs/2508.06108v1. Although the details differ, the general direction and motivation are highly similar. The authors should clarify the distinctive technical contribution and explain how their algorithm fundamentally advances beyond these existing efforts. Additionally, as noted earlier, the maze environment is simple enough that Hindsight Experience Replay alone can achieve strong performance. It is unclear whether a full hierarchical setup is necessary for such a domain.

---

> ### Author Response · Authors · 2025-11-28
>
> Thank you for your comments. We address your concerns point-by-point:
> 1. **Limitation of novelty** Separating low- and high-level policies is not the main idea of our work - this is common to all hierarchical GCRL algorithms. Our contribution, as clarified in the general comment and new related work, is 1) the pre-training of a goal-conditioned low-level policy and 2) a transfer-based model (with a novel architecture) which explicitly learns reachability from data, improving landmark generation. Neither of these have been done before in the field (previous methods use simplistic reachability heuristics and train levels jointly). Our method indeed tackles the fundamental bottleneck in our field of GCRL. The bottleneck is not related to domain knowledge, but instead it is the sample efficiency problem in long-horizon planning (most GCRL papers introduce the research gap in this way). The high-level planner can reason about long-term transitions since it is a shortest path algorithm over a graph of landmarks. This graph-based discretisation is the main feature of all graph-based hierarchical GCRL methods (our contribution is fewer landmarks for faster exploration). The trajectory model and low-level policy only need to reason about the single next landmark/sub-goal, so they can be trained in a much smaller/simpler environment.
> 2. **The design of LSTM** LSTMs are neural models and, as such, are ideal for high-dimensional scenarios. They have been used extensively in natural language processing, given inputs in high-dimensional embedding spaces. They can even handle long sequence lengths, though this capability is not needed for our algorithm since the trajectory model is trained in the short-horizon, simple environment. As for the Reacher experiments, we already performed detailed statistical and visualisation investigations in the original submission providing strong evidence that the accessibility model was to blame for the degraded results rather than the trajectory model (see Appendix section H in the new revision). Indeed, the use of the neural density model improved the results in Reacher-wall (section 5.2), thereby vindicating the LSTM trajectory model.
> 3. **Weak baselines and missing analysis** NGTE, BEAG and DHRL are not weak baselines, they are the state of the art in hierarchical GCRL. They are much newer and more performant than SoRB or CO-PILOT. DHRL in particular samples from the replay buffer and uses the value function as a reachability heuristic. SoRB is the first graph-based GCRL algorithm of many and uses all states in the buffer as nodes (very inefficient). It does not plan in goal space, but in state space and therefore cannot handle long-horizon tasks (and is not tested on these). DHRL is a much better version of this since it uses farthest point sampling on the buffer rather than all states and plans in goal space (even if the initial state distribution is fixed instead of uniform). Yet DHRL could not solve the AntMaze tasks at all, demonstrating that SoRB would have no hope. CO-PILOT similarly does not test on mazes anywhere near the size of those tested in the SOTA baselines against which we compared. It is very unclear why SoRB/CO-PILOT have been singled out here as they are just standard old GCRL algorithms which do not compete with the SOTA. Additionally, we already provided detailed wall clock time results in the original submission, demonstrating that there is no additional computational burden per environment step for pre-training. See the ‘Computation’ subsection of the results and the ‘Wall Clock Time’ Appendix section.
> 4. **Questions** The linked paper seems mostly irrelevant to ours. It is not graph-based, does not generate landmarks, does not test on long-horizon tasks, does not pre-train or transfer anything and is unpublished. Could the reviewer please clarify the relevance here? Additionally, this statement is categorically false: “AntMaze is simple enough that HER alone can achieve strong performance”. Could you please clarify what results this is based on? Even in older papers, e.g. HAC/LEAP/etc, HER very clearly could not solve even the most basic AntMaze tasks, let alone the very large and complex mazes in SOTA papers. The AntMaze suite allows for very complex, long-horizon tasks (and has a 31D state space) - that is why most GCRL papers use it as the main testbed. We have followed the experimental protocol in our field closely. As for novelty, we have now clarified this in the general comment and the new related work section.

---

### Official Review · Reviewer_as96 · 2025-11-03

**Soundness:** 2
**Presentation:** 2
**Contribution:** 2
**Rating:** 2
**Confidence:** 4

**Summary:**

This paper introduces PROMO (Progressive Reachability Optimisation MOdelling), a novel approach to improve sample efficiency in Goal-Conditioned Reinforcement Learning (GCRL). The key innovation is the application of policy transfer to GCRL by pre-training a low-level policy in a simple, obstacle-free environment and then transferring it to complex target environments. The method consists of three main components: (1) a transferable low-level GCRL policy trained using TD3+HER, (2) a trajectory model that predicts the policy's behavior, and (3) a high-level graph-based planner that uses a smart landmark generation procedure. The landmark generation optimizes over novelty, reachability, and boundary separation scores to progressively cover the goal space. Experiments on AntMaze and Reacher environments demonstrate 3-4x improvements in sample efficiency and near 100% success rates compared to state-of-the-art methods.

**Strengths:**

1. This paper introduces a new landmark generation procedure, incorporating three meaningful components (novelty access, reachability, and boundary separation) that lead to fewer but more informative landmarks.

**Weaknesses:**

1. The novelty and motivation of this paper is not clear. The authors claim their contribution as pretrained transfer skills in GCRL. However, a general game AI model should be tested beyond the navigation tasks (e.g., chess, shogi, Atari).
2. The novelty of algorithms is very limited. There are many previous GCRL measured the reachability of the low-level policy.
3. Moreover, the method is constrained by the choice of SVM ensemble for accessibility modeling, which limits applicability to (i) Low-dimensional goal spaces only (2D for AntMaze, 3D for Reacher) and (ii) Environments where obstacles can be modeled by the SVM ensemble.

**Questions:**

1. How sensitive is the method to the choice of a simple environment? What guidelines exist for constructing appropriate, simple environments for different domains?
2. How robust is the method when there are significant differences in dynamics between the simple and complex environments beyond just obstacle placement?
3. How could the method be extended to higher-dimensional goal spaces? Would replacing the SVM ensemble with neural density models (as suggested) maintain the same performance benefits?
4. Could the pre-trained policy transfer across different types of environments (e.g., from navigation to manipulation) or is it limited to variations within the same domain?
5. How would the method perform in real robotic systems where there might be additional sources of uncertainty and dynamics mismatch?

---

> ### Author Response · Authors · 2025-11-28
>
> Thank you for your comments. We address your concerns point-by-point:
> 1. **Novelty + problem space** We have now clarified the novelty in the general comment and new related work as not only the transfer of a goal-conditioned low-level policy (which is new), but the use of this to explicitly model reachability for better landmark generation. It is unclear what is meant by ‘general game AI’ - our method is a GCRL algorithm, meaning that it solves goal-reaching tasks (goal-augmented MDPs), an established sub-field within RL. Chess/shogi/Atari are not goal-reaching tasks and are not applicable. GCRL algorithms typically test on continuous, sparse reward navigation and manipulation tasks, especially AntMaze and Reacher. We have followed our field’s established experimental protocol closely.
> 2. **Novelty of measuring reachability** This has now been clarified. Yes, every landmark generation method in hierarchical GCRL implicitly or explicitly measures reachability, but does not learn it from data, instead using simplistic heuristics which are inaccurate for far-apart goals (see new related work). Ours is the first to learn reachability (and novelty) from data, even for far-apart goals. This allows better landmark generation.
> 3. **Dimensionality** This point has now been fully addressed in the general comment and section 5.2 of the new revision.
> 4. **Constructing simple envs** This point was already discussed in section 4.1. We presented three desiderata for the simple environment and argued that this is easy and intuitive to construct. In the worst case, the main environment can be used with goals limited to a small vicinity of the initial achieved goal.
> 5. **Differences in LL/HL dynamics** Since one can construct a simple pre-training environment, there is no need to adapt policies trained for very different tasks. Nevertheless, our third ablative study tested mismatched dynamics between the levels by adding action noise to the low-level policy. It showed good robustness to the mismatch.
> 6. **Neural density model** These results have been added to section 5.2, showing good performance. This was a proof-of-concept but we expect future work to find a more appropriate neural method and ensembling techniques.
> 7. **Cross-domain transfer** Cross-domain transfer of a low-level policy is impossible since the state/action spaces fundamentally mean different things. Even meta-RL and skills based HRL methods transfer within-domain (e.g. the MetaWorld benchmark). This still provides a massive transfer potential, e.g. transfer to any size/topology of maze.
> 8. **Uncertainty / real-life domains** Our third ablative study showed robustness to uncertainty/noise. We are interested in the future possibility of transferring a low-level policy trained with a simulated robot arm to a real environment for high-level planning (not possible with current jointly trained methods). However, testing on real hardware is out-of-scope in this paper and, more generally, not common in the GCRL literature.

---

### Author Response · Authors · 2025-11-28
**General Comment**

We thank all reviewers for their detailed and insightful comments, based on which we have made significant updates to the submission. Changes in the new revision are highlighted in blue. Here is a summary of the main changes:

1. **Clarification of novelty** Existing methods in hierarchical GCRL all balance reachability with novelty using overly simplistic heuristics, which are not accurate for far-apart goals in the goal space, leading to the need for many close-by landmarks and slow exploratory expansion through the goal space. To remedy this, we explicitly model reachability between any two goals (using a transferred low-level policy) and use this to generate landmarks which are significantly fewer and more spaced out while still being reachable. The reachability model has a novel trajectory + accessibility architecture. Our work is the first in GCRL to model reachability explicitly in a way that is accurate even for far-apart goals. It is also the first to demonstrate the benefits of transferring a low-level policy in GCRL. A detailed discussion has been added to the Related Work section.
2. **Transferred policy in baselines** To prove that our results were not just due to transferring the low-level policy in a way which could be done for the baselines, we tried to transfer our pre-trained policy to the best performing baselines. However, they were not able to utilise the policy (e.g. BEAG only managed to explore half of Zonal-maze using it), showing that it is the combination of the transferred policy and the reachability model / landmark generation procedure that leads to the efficiency and success improvements. These results and a discussion have been added to section 5.1 (Comparative Results) of the new revision.
3. **High dimensions** First, we clarify that GCRL methods either tackle simple, short-horizon tasks with complex high-dimensional (visual) state/goal spaces, or they tackle complex long-horizon tasks (like AntMaze) with low-dimensional goal spaces. Since ours falls into the latter category (which also includes the vast majority of GCRL methods), high-dimensional testing is out-of-scope for our work, as also in the SOTA baseline papers. Nevertheless, we seek to address the reviewers’ concerns by showing that the SVM ensemble can be replaced by a neural method. As a proof-of-concept, we ran the algorithm with a normalising flow density estimator in place of SVM for U-maze and Reacher-wall, showing good results. Although more work is needed to find the most appropriate neural method, this suggests that there is nothing stopping our method from working in high dimensions. The results and discussion have been added to section 5.2.
4. **Number of landmarks** We have added visual examples demonstrating that our method produces far fewer landmarks than the baselines (section 5.2 and Appendix G)
5. We also address the reviewers’ individual concerns below. Please let us know to what extent these additions have satisfied your concerns, and what else you would like to see.

---

### Note · Authors · 2026-01-23

I have read and agree with the venue's withdrawal policy on behalf of myself and my co-authors.